# Dietary Leucine Improves Fish Intestinal Barrier Function by Increasing Humoral Immunity, Antioxidant Capacity, and Tight Junction

**DOI:** 10.3390/ijms24054716

**Published:** 2023-03-01

**Authors:** Ju Zhao, Ye Zhao, Haifeng Liu, Quanquan Cao, Lin Feng, Zhihao Zhang, Weidan Jiang, Pei Wu, Yang Liu, Wei Luo, Xiaoli Huang, Jun Jiang

**Affiliations:** 1College of Animal Science and Technology, Sichuan Agricultural University, Chengdu 611130, China; 2Animal Nutrition Institute, Sichuan Agricultural University, Chengdu 611130, China; 3Fish Nutrition and Safety Production University Key Laboratory of Sichuan Province, Sichuan Agricultural University, Yaan 625014, China

**Keywords:** leucine, intestinal barrier function, humoral immunity, antioxidant capacity, tight junction, autophagy

## Abstract

This study attempted to evaluate the possible impact and mechanism of leucine (Leu) on fish intestinal barrier function. One hundred and five hybrid *Pelteobagrus vachelli* ♀ × *Leiocassis longirostris* ♂ catfish were fed with six diets in graded levels of Leu 10.0 (control group), 15.0, 20.0, 25.0, 30.0, 35.0, and 40.0 g/kg diet for 56 days. Results showed that the intestinal activities of LZM, ACP, and AKP and contents of C3, C4, and IgM had positive linear and/or quadratic responses to dietary Leu levels. The mRNA expressions of itnl1, itnl2, c-LZM, g-LZM, and β-defensin increased linearly and/or quadratically (*p* < 0.05). The ROS, PC, and MDA contents had a negative linear and/or quadratic response, but GSH content and ASA, AHR, T-SOD, and GR activities had positive quadratic responses to dietary Leu levels (*p* < 0.05). No significant differences on the CAT and GPX activities were detected among treatments (*p* > 0.05). Increasing dietary Leu level linearly and/or quadratically increased the mRNA expressions of CuZnSOD, CAT, and GPX1α. The GST mRNA expression decreased linearly while the GCLC and Nrf2 mRNA expressions were not significantly affected by different dietary Leu levels. The Nrf2 protein level quadratically increased, whereas the Keap1 mRNA expression and protein level decreased quadratically (*p* < 0.05). The translational levels of ZO-1 and occludin increased linearly. No significant differences were indicated in Claudin-2 mRNA expression and protein level. The transcriptional levels of Beclin1, ULK1b, ATG5, ATG7, ATG9a, ATG4b, LC3b, and P62 and translational levels of ULK1, LC3Ⅱ/Ⅰ, and P62 linearly and quadratically decreased. The Beclin1 protein level was quadratically decreased with increasing dietary Leu levels. These results suggested that dietary Leu could improve fish intestinal barrier function by increasing humoral immunity, antioxidative capacities, and tight junction protein levels.

## 1. Introduction

The intestine plays an important role in nutrient absorption and it is a key immunological barrier and physical barrier to the entry of harmful substances [1]. Amino acids play an important regulating role in intestinal barrier function [2]. Leucine (Leu) is one of the branched-chain amino acids, which are essential for human, fish, and other animal species [3,4]. Supplementation of dietary Leu or providing a Leu-rich diet will increase the protein accretion in tissues [3,5]. Yellow catfish (*Pelteobagrus fulvidraco*) is a common and commercially significant freshwater fish species raised in China. Hybrid catfish have been developed in recent years via breeding female *Pelteobagrus vachelli* and male *Leiocassis longirostris*. The hybrid catfish showed improved traits compared to their parents such as better growth performance and hypoxia tolerance [6]. Previous reports mainly focused on lipid accumulation, oxidative stress [7], disease resistance [8], and water temperature and stocking density [6]. Our earlier research determined optimal dietary tryptophan [9], threonine [10], and isoleucine [11] levels. In our previous study, the Leu requirement of hybrid catfish (23.19–54.55 g) for percent weight gain was estimated to be 28.10 g/kg of the diet (73.04 g/kg of dietary protein) based on the broken-line model [12]. However, no study has investigated the effect of Leu on intestinal barrier function in this fish.

The intestinal immune barrier in fish is an important component of the intestinal barrier [13], which is maintained via humoral immune factors, such as lysozyme (LZM) [14,15,16], acid phosphatase (ACP), alkaline phosphatase (AKP) [11], complements (C3 and C4), immunoglobulin M (IgM) [17,18], and antimicrobial peptides (hepcidin and β-defensin) [11,19]. Previous studies have reported that Leu increased LZM activity and C3 and C4 contents in the intestine of juvenile golden pompano (*Trachinotus ovatus*) [20], and C3 and IgM contents in the liver of juvenile blunt snout bream (*Megalobrama amblycephala*) [21]. These data suggested that dietary Leu could enhance the intestinal immune barrier through increased humoral immunity. However, little research has focused on the influence of Leu on the intestinal immune barrier in hybrid catfish, which deserves further investigation.

The reactive oxygen species (ROS) frequently found to metabolic production and fodder environments [22,23], increase the chance of oxidative damage and disturbed intestinal physical barrier function [24]. Antioxidant capacity is important for maintaining intestinal physical barrier function [13]. Fish intestinal antioxidant capacity is improved with increasing antioxidant enzymes activities, non-enzymatic antioxidant content, and up-regulating gene expressions of antioxidant enzymes [25]. The Nrf2/Keap1 signaling pathway is a fundamental signaling pathway responsible for regulating gene expressions of antioxidant enzymes [26]. Previous studies have indicated that Leu enhanced antioxidant enzyme activities and non-enzymatic antioxidant content through improving corresponding antioxidant gene expressions via the Nrf2/Keap1 signaling pathway in liver of the juvenile blunt snout bream [21] and intestine of young grass carp (*Ctenopharyngodon idella*) [27]. However, it is still unclear what impact Leu has on the intestinal antioxidant capacity of hybrid catfish.

Tight junctions (TJs) play an important role in paracellular barrier functions. TJs comprise transmembrane proteins, such as occludin and Claudins, and intracellular plaque proteins, such as ZO-1 [28]. The impairment of intestinal TJ protein expression leads to the disruption of paracellular barrier function [29]. A previous study indicated that optimal dietary Leu increases the mRNA levels of occludin and ZO-1 in the intestine of grass carp [30]. Leu significantly enhanced protein levels of occludin in human colon carcinoma cell line LS174T [31] and increased the Claudin-2 protein level in human Caco2 BBe cells [32]. These up-to-date data suggested that Leu may enhance intestinal TJs by promoting protein expression of TJs via increasing their mRNA levels. Previous studies indicated that autophagy inhibition can significantly increase the protein level of occludin in mouse brain endothelial cells [33] and the protein level of ZO-1 rat brain microvascular endothelial cells [34]. A limited study indicated that Leu inhibits autophagy in zebrafish sperm [35]. However, whether Leu is involved in regulating intestinal TJs by inhibiting autophagy merits further investigation.

Therefore, the objectives of the present study are to investigate the possible impacts and potential mechanisms of dietary Leu on the intestinal barrier function in hybrid catfish. These results may provide a partial theoretical basis to reveal the potential regulatory approach for intestinal barrier function by Leu in fish.

## 2. Results

### 2.1. The Humoral Immune-Related Parameters in Intestine

The orthogonal polynomial contrasts showed that there were significant interactions of intestinal LZM, ACP and AKP activities, and C3, C4, and IgM contents, with graded dietary Leu levels (Table 1). The increasing Leu levels led to significant linear and quadratic increases in intestinal LZM, ACP, and AKP activities and C4 content (*p* < 0.05). With increasing dietary Leu level, the C3 content increased linearly (*p* < 0.05). The IgM content increased quadratically (*p* < 0.05). The itnl1, itnl2, c-LZM, and g-LZM mRNA expression increased linearly and quadratically (Figure 1A–D, *p* < 0.05). The β-defensin mRNA expression increased quadratically with increasing dietary Leu levels (Figure 1E, *p* < 0.05). The above results suggested that dietary Leu enhanced the intestinal immune barrier by increasing humoral immune-related parameter expression.

### 2.2. Antioxidant-Related Parameters in Intestine

The orthogonal polynomial contrasts showed that there were significant interactions of intestinal PC, MDA, ROS, and GSH contents, and ASA, AHR, T-SOD, CAT, GPX, GST, and GR activities, with graded dietary Leu levels (Table 2). The ROS, PC, and MDA contents were linearly and/or quadratically decreased with increasing dietary Leu levels (*p* < 0.05). No significant differences in the CAT and GPX activities were detected among treatments (*p* > 0.05). Increasing dietary Leu levels quadratically increased the GSH content, and ASA, AHR, T-SOD, and GR activities (*p* < 0.05). The GST activity was linearly increased with increasing dietary Leu levels (*p* < 0.05).

The CuZnSOD mRNA expression increased quadratically (Figure 2A, *p* < 0.05). With increasing dietary Leu levels, the CAT and GPX1α mRNA expressions increased linearly (Figure 2B,D, *p* < 0.05). The GST mRNA expression decreased linearly (Figure 2C, *p* < 0.05). The GCLC and Nrf2 mRNA expressions were not significantly affected by different dietary Leu levels (Figure 2E,F, *p* > 0.05). The Nrf2 protein level quadratically increased (Figure 2H, *p* < 0.05). With increasing dietary Leu level, Keap1 mRNA expression and protein level decreased quadratically (Figure 2G,I, *p* < 0.05). The results demonstrated that dietary Leu increased antioxidant gene expression via the Nrf2/Keap1 signaling pathway.

### 2.3. Tight Junction Protein and Autophagy-Related Parameters in Intestine

Immunochemistry analysis of TJ proteins, occludin, and ZO-1 showed that they were highly expressed in the fish fed the 25.0 g Leu/kg diet (Figure 3A). There was a linear and quadratic effect of dietary Leu levels on occludin mRNA expression (Figure 3B, *p* < 0.05). With increasing dietary Leu level, ZO-1 mRNA expression and protein level and occludin protein level significantly increased quadratically (Figure 3C–E, *p* < 0.05). Dietary Leu levels had no significant difference on Claudin-2 mRNA expressions and protein level (Figure 3F,G, *p* > 0.05). The orthogonal polynomial contrasts showed that the increasing Leu levels linearly and quadratically decreased the Beclin1, ULK1b, ATG5, ATG7, ATG9a, ATG4b, and LC3b mRNA expressions and ULK1 and LC3II/I protein levels, and linearly and quadratically significantly increased mRNA expression and protein level of P62 (Figure 4A,B,D–L, *p* < 0.05). The Beclin1 protein level was quadratically decreased with increasing dietary Leu levels (Figure 4C, *p* < 0.05). These results suggested that dietary Leu improved intestinal TJ function via up-regulating occludin and ZO-1 expressions and down-regulating autophagy-related parameter expressions.

## 3. Discussion

### 3.1. Dietary Leu Improved Immune Barrier Function in the Intestine of Fish

The immune function of the intestine is also an important element of the intestinal barrier [11]. Immune parameters such as immune-related enzymes (LZM and ACP), complement, and immune globulin have been regarded as crucial tools for investigating the intestinal immune barrier in fish [36,37]. LZM, as a first barrier against microbial invasion, participates in the innate immune response in fish [38,39,40,41]. ACP and AKP, important hydrolytic enzymes in lysosomes, have a beneficial effect on defense of the body against foreign pathogens and microorganism invasion [42]. The central components of the complement system mainly include C3 and C4, which play an essential role in alerting about potential pathogens in the host [43]. IgM has been well characterized in fish and seems to be specialized in systemic immunity [44]. The present study therefore further detected the effect of humoral immunity in hybrid catfish intestine and found the activities of LZM, ACP, and AKP and C3, C4, and IgM contents had positive linear and/or quadratic responses to dietary Leu levels. These results were in agreement with previous findings in the intestine of grass carp [30], and head kidney of *Labeo rohita* fingerlings [45]. Intelectin is a glycan-binding lectin, and it plays a major role in bacterial agglutination and binding capacity, as well as polysaccharide recognition in blunt snout bream [46]. The LZM activity was regulated by c-type and g-type LZM, which catalyzed the hydrolysis of bacterial cell walls [11,47]. β-defensins have been considered as a major class of antibacterial peptides, which have significant effect on congenital immunity of bony fish against bacteria [48,49]. In the present study, the itnl1, itnl2, c-LZM, g-LZM, and β-defensin mRNA expression had a positive linear and/or quadratic response in the intestine of hybrid catfish. These results are parallel to the previous studies showing that Leu dramatically improved mRNA expression of β-defensin in the intestine of weaned piglets and IPEC-J2 [50]. These results implied that dietary Leu exerts a positive effect on intestinal immune barrier function in hybrid catfish. A fish’s intestinal microbiota is essential to the host because it controls metabolic function, pathogen resistance, immunological activity, and feed conversion [51,52]. Previous research indicates that the intestinal microbiota may act as a mediating factor in the positive effect of dietary Leu supplementation on intestinal health of mice [53]. It is also unknown whether dietary Leu regulates the function of the intestinal immune barrier by affecting intestinal microbiota in hybrid catfish, which needs further study.

### 3.2. Dietary Leu Enhanced Physical Barrier Function via Up-Regulating Intestinal Antioxidant Capacity in the Intestine of Fish

The physical barrier function of the fish intestine is the first line of defense against infection [54], which is concerned with the cellular antioxidative state [11,55]. MDA and PC are considered to be important biochemical indicators, which are usually used to reflect lipid peroxidation and protein oxidation in tissues [56,57]. Due to the inescapable exposure to foreign materials, the intestine is also considered as a critical source of ROS [58]. Insufficient scavenging of ROS contributes to oxidative injury in the intestine [57]. This result demonstrated that dietary Leu linearly and/or quadratically decreased ROS, PC, and MDA contents in the intestine of hybrid catfish. Correlation analysis found that ROS was positively correlated with MDA (r = +0.912, *p* = 0.004) and PC contents (r = +0.793, *p* = 0.033) (Table 3), which indicated that dietary optimal Leu could ameliorate oxidative damage by reducing ROS accumulation. Studies on the gills of grass carp and porcine intestinal epithelial cells also demonstrated that Leu reduced the level of cell ROS [57]. The activities of ASA and AHR are two vital indices to assess the ability to scavenge superoxide anions (O^2−^) and hydroxyl radicals (OH^−^) in the intestine and hepatopancreas of juvenile Jian carp (*Cyprinus carpio var. Jian*) [59]. Here, we found that the activities of ASA and AHR had positive quadratic responses to dietary Leu levels. Correlation analysis showed that the ROS content was negatively related to ASA (r = −0.956, *p* = 0.001) and AHR (r = −0.954, *p* = 0.001) activities in the intestine of hybrid catfish (Table 3), which suggested that dietary Leu might decrease ROS accumulation by enhancing ASA and AHR activities. Previous researchers also reported that dietary Leu enhanced the activities of ASA and AHR in the gills of grass carp [60].

To alleviate oxidative damage caused by ROS, fish have formed efficient antioxidant networks that can be divided into two categories, non-enzymatic antioxidants (such as GSH) and antioxidant enzymes (such as CuZnSOD, CAT, GPX, and GR) [61,62]. The present experimental results indicated that GSH content and T-SOD, GST, and GR activities were linearly and/or quadratically increased with increasing dietary Leu levels. No significant differences in the CAT and GPX activities were detected among treatments. Correlation analysis showed that ROS was negatively correlated with GSH (r = −0.965, *p* = 0.000), T-SOD (r = −0.933, *p* = 0.002), CAT (r = −0.928, *p* = 0.003), GPX (r = −0.920, *p* = 0.003), GST (r = −0.943, *p* = 0.001), and GR (r = −0.942, *p* = 0.001) (Table 3), showing that Leu might diminish the accumulation of ROS via improving the intestinal GSH content, and T-SOD, CAT, GPX, GST, and GR activities. These results were in accordance with previous reports in the muscle of grass carp and intestine of juvenile golden pompano [20,27,63]. Increases in GSH content may be attributed to Leu metabolism. Leu can be transamined to form glutamate, which can be used as a source for mucous GSH synthesis [64,65]. The enzyme activities of antioxidants show positive relationships with their gene expressions in fish [66,67]. The current study found that the CuZnSOD, CAT, and GPX1α mRNA expressions were linearly and/or quadratically significantly increased with increasing dietary Leu levels. The GST mRNA expression decreased linearly while the GCLC mRNA expression was not significantly affected. Further correlation analyses showed that CuZnSOD, CAT, and GPX1α were, respectively, positively correlated with T-SOD (r = 0.981, *p* = 0.00), CAT (r = 0.861, *p* = 0.013), and GPX (r = 0.643, *p* = 0.119) (Table 3), which is in agreement with the results of previous research on the intestine of Jian carp [68]. Transcription factor Nrf2 was regarded as a pivotal regulator of the cellular antioxidant response. Keap1 is a critical sensor in cellular oxidative stress, which can bind to Nrf2 and constantly promotes its degradation, and is a negative regulator to switch off the Nrf2 response [26,69]. In this study, we observed that the Nrf2 mRNA expression was not significantly affected by different dietary Leu levels. The Nrf2 protein level quadratically increased, whereas the Keap1 mRNA expression and protein level decreased quadratically in the intestine of hybrid catfish. Correlation analysis showed that mRNA expression of Nrf2 was positively correlated with CuZnSOD (r = +0.892, *p* = 0.007), CAT (r = +0.794, *p* = 0.033), GPX1α (r = +0.849, *p* = 0.008), and GCLC (r = +0.818, *p* = 0.025) mRNA expressions, and Keap1 mRNA expression was negatively correlated with CuZnSOD (r = −0.975, *p* = 0.000), CAT (r = −0.906, *p* = 0.005), GPX1α (r = −0.794, *p* = 0.033), and GCLC (r = −0.919, *p* = 0.003) mRNA expressions (Table 3). Western blot analysis demonstrated that dietary Leu increased the protein level of Nrf2 and reduced the protein level of Keap1. Similar results were observed in the intestine of juvenile golden pompano [20] and the head kidney of *Labeo rohita* fingerlings [45]. Taken together, these results clearly indicated that dietary Leu could enhance intestinal antioxidative capacity by modulating the Nrf2/Keap1 signaling pathway in hybrid catfish.

### 3.3. Dietary Leu Enhances Paracellular Barrier Functions via Increasing TJ Protein Levels in the Intestine of Fish

The paracellular barrier function of the intestine is related to the level of TJ proteins between epithelial cells [70]. The TJ proteins act as a paracellular barrier and serve as a first line of cellular defense against paracellular permeation of noxious luminal antigens [71,72]. Occludin, ZO-1, and Claudin-2, as the major components of TJ proteins, affect intestinal paracellular barrier functions [73]. Occludin plays a critical role in formation of TJ seals, and its damage facilitates macromolecule flux across the intestinal epithelial barrier [74]. ZO-1 is a cytoplasmic plaque protein that interacts with both transmembrane proteins and cytoskeletal proteins [75]. In the present study, the mRNA expression and protein level of intestinal ZO-1 and protein level of occludin significantly increased quadratically. Similarly, a study on the intestine of grass carp showed that optimal dietary Leu up-regulated occludin and ZO-1 mRNA expressions [30]. An in vitro model of intestinal epithelium lines demonstrated that Leu significantly enhanced occludin protein production [31]. Claudin-2 is expressed in the TJs of leaky epithelia, which is responsible for the flux of cations and small solutes [76]. In the current study, no significant differences were found in Claudin-2 mRNA expression and protein level. Claudin-2 knockout can cause defects in paracellular Na^+^ flow and nutrient transport in the intestine and result in death from malnutrition [77]. These results were in agreement with our previous report. Optimal dietary Leu improved feed efficiency in hybrid catfish [12] (Appendix A). However, the underlying mechanism needs further investigation. The present study is the first to show that dietary Leu up-regulated the TJ function, partly ascribed to increases in TJ protein levels. New evidence revealed that autophagy plays an important role in maintaining occludin and ZO-1 protein levels [78]. The autophagy process involves three major phases: autophagosome initiation, elongation, and completion [79]. Beclin1 and ULK1 are involved in autophagy initiation [80,81]. ATG5 initiates the formation of double membrane vesicles [82]. Autophagy requires ubiquitin-like ATG8 and ATG12 conjugation systems, where ATG7 has a critical role as the sole E1 enzyme [83]. ATG9 is a multispanning membrane protein, which is essential for autophagy [84]. ATG4 is a key cysteine protease, which is crucial for proper biogenesis of the autophagosome [85]. The marker protein of autophagy is microtubule-associated protein light-chain 3 (LC3), which is responsible for the fusion of autophagosomes to lysosomes and formation of autolysosomes [86]. P62 is a multifunctional, cytoplasmic protein, which is degraded by autophagy in autophagy-mediated degradation progresses [87]. The present study observed that the Beclin1, ULK1b, ATG5, ATG7, ATG9a, ATG4b, and LC3b mRNA expressions and ULK1, Beclin1, and LC3II/I protein levels linearly and/or quadratically decreased, and mRNA expressions and protein levels of P62 linearly and quadratically increased, with increasing dietary Leu levels. A similar observation in sperm of zebrafish revealed that Leu suppressed autophagy by inhibiting the fusion of autophagosomes and lysosomes [35]. In addition, related research in HeLa cells showed that Leu alleviated autophagy via the impact of its metabolite AcCoA on mTORC1 [88]. Therefore, Leu might down-regulate autophagy levels in the fish intestine. Autophagy regulates the level of TJ proteins and therefore changes epithelial barrier function [89]. Di-(2-ethylhexyl) phthalate exposure destroyed the blood–testis barrier integrity of the immature testis through excessive ROS-mediated autophagy [90]. Autophagy inhibition increased occludin protein level in mouse brain endothelial cells [33]. In human squamous cell carcinoma cells, inhibition of autophagy can up-regulate ZO-1 protein level [91]. A previous study has demonstrated that Claudin-2 enhances the intestinal permeability [89]. Studies in MDCK I cells have shown that Claudin-2 mediated cation and water transport [92]. Thus, Claudin-2 may be beneficial for nutrient absorption of small molecules, which needs further investigation. These findings might partly explain why dietary Leu increased occludin and ZO-1 protein levels through down-regulating autophagy. Nevertheless, the detailed mechanism still requires a further investigation.

## 4. Materials and Methods

### 4.1. Experimental Diets, Feeding Trial, and Sampling

This present experiment used a total of 630 fish with an average initial weight of 23.19 ± 0.20 g were randomly distributed into 21 tanks, with 30 fish in each tank, and feed formulation according to our previous study [12] (Appendix A). The Leu was added to the experimental diets at levels of 10.0 (control), 15.0, 20.0, 25.0, 30.0, 35.0, and 40.0 g/kg diet [12] (Appendix A). After the 8-week feeding trial, twelve fish from each replicate were anesthetized with benzocaine solution (50 mg/L). Then, fish were sacrificed on the basis of the methods of Lisbeth et al. [93]. The intestine samples were obtained and stored at −80 °C until further analysis.

### 4.2. Biochemical Measurements

The samples of intestine were carefully homogenized with 10 volumes (mg/mL) of chilled physiological saline solution. The homogenates were centrifuged at 6000 g for 20 min at 4 °C. The supernatant was collected for antioxidant and humoral immune indicators. The MDA, PC, and GSH contents and ASA, AHR, T-SOD, CAT, GPX, GST, and GR activities were determined as described by Jiang et al. [55]. The LZM, AKP, and ACP activities, and C3, C4, and IgM contents, were measured as described in the study by Zhao et al. [11]. The ROS content was spectrophotometrically assayed by detecting the oxidation of 2′,7′-dichlorofluorescein [94].

### 4.3. Immunohistochemistry (IHC) Staining

The intestine tissues were embedded in paraffin and sectioned into slices 6 μm thick. Slices were incubated at 37 °C for 24 h before being incubated at 60 °C overnight. Following normal immunohistochemistry procedures, paraffin-embedded tissue slides were deparaffinized with xylene, and dehydrated with gradient ethanol. Heat-induced antigen retrieval was carried out for 15 min at 95 °C in citrate buffer [95]. This was followed by a 15 min incubation with 3% hydrogen peroxide at room temperature for elimination of endogenous peroxidase. Next, the tissue sections were blocked with 2% bovine serum albumin (BSA) for 30 min at room temperature, and then incubated overnight at 4 °C with occludin and ZO-1 polyclonal antibodies (diluted 1:100 with TBST; ABclonal, Chengdu, China). Afterwards, the tissue sections were incubated for 15 min at 37 °C with IgG secondary antibody, and then for another 30 min with peroxidase-labeled streptomycin (MXB Biotechnologies, Fuzhou, China), followed by washing for 5 min with phosphate buffer thrice. The sections were visualized with a DAB chromogenic substrate kit, and the nuclei were stained with hematoxylin. Imaging and analysis were carried out under an Olympus BX43F upright microscope.

### 4.4. Quantitative Real-Time PCR (qRT-PCR)

Total RNA from intestinal tissues was extracted according to the manual of the RNAiso Plus kit (TaKaRa, Dalian, China), and quantified by DU 640 UV spectrophotometer detection (Beckman, USA) at 260 and 280 nm. Using 1.0% agarose gels detecting the integrity of RNA, thereafter, we immediately reverse-transcribed it into cDNA by using the PrimeScript^®^ RT reagent kit with gDNA Eraser (TaKaRa, Dalian, China). Sequences of primers used to perform RT-qPCR are listed in Appendix A. The qRT-PCR analysis was performed using a CFX96 Real-Time PCR Detection System (Bio-Rad, Hercules, CA, USA). The relative mRNA expression was computed using the 2^−ΔΔCT^ method, and the final results were all normalized to house-keeping genes (including β-actin and 18S rRNA).

### 4.5. Western Blot Analysis

The protein from intestines was extracted according to a previous study [12]. Protein concentration was determined using a protein quantification kit (Beyotime, Shanghai, China). The protein samples were subjected to SDS-PAGE and then transferred onto a PVDF membrane (Millipore, Inc., Bedford, MA) by wet electroblotting. After blocking with 5% bovine serum albumin (in Tris-buffered saline containing Tween 20) at room temperature for approximately 2 h, samples were subsequently incubated overnight in primary antibodies: Nrf2 and Keap1 (1:2000, Zen Biotechnology, Chengdu, China), Beclin1, ULK1, LC3Ⅱ/Ⅰ, P62, occludin, ZO-1, and Claudin-2, (1:2000, ABclonal, Chengdu, China). The membranes were subjected to washing three times, and then incubated with horseradish peroxidase (HRP)-conjugated secondary antibodies for approximately 2 h. β-actin (1:1000; CST) and Lamin B1 (1:2000, Zen Biotechnology) were considered as the control proteins for total and nuclear protein. The immune response bands were measured by enhanced chemi-luminescence. Quantify protein expression was detected by a Gel-Pro Analyzer (Media Cybernetics Bethesda, MD, USA), and analyzed by the ImageJ gel analysis software [96].

### 4.6. Statistical Analysis

The trends of linear and quadratic analyses were measured by using SAS software version 8.0 (SAS Institute Inc., Cary, NC, USA) [97,98]. All results were subjected to one-way ANOVA analysis followed by Duncan’s multiple range test to evaluate the differences within treatments using SPSS version 25.0 (SPSS Institute Inc., Chicago, IL, USA). The level of significance for all analyses was considered *p* < 0.05. Data were expressed as the mean ± SEM. Bivariate correlation analysis was performed with Pearson correlation.

## 5. Conclusions

In summary (Figure 5), dietary Leu enhanced intestinal immune barrier function via increasing humoral immune factors and related genes’ expression, and intestinal physical barrier function by regulating intestinal antioxidant capacity via the Nrf2/Keap1 signaling pathway, in fish. In addition, this study provides evidence that dietary Leu improved the functioning of intestinal TJ by down-regulating autophagy in fish, but the detailed mechanism needs further study. These results of the study will fill the knowledge gaps regarding the impact of Leu on fish intestinal barrier function.

## Figures and Tables

**Figure 1 ijms-24-04716-f001:**
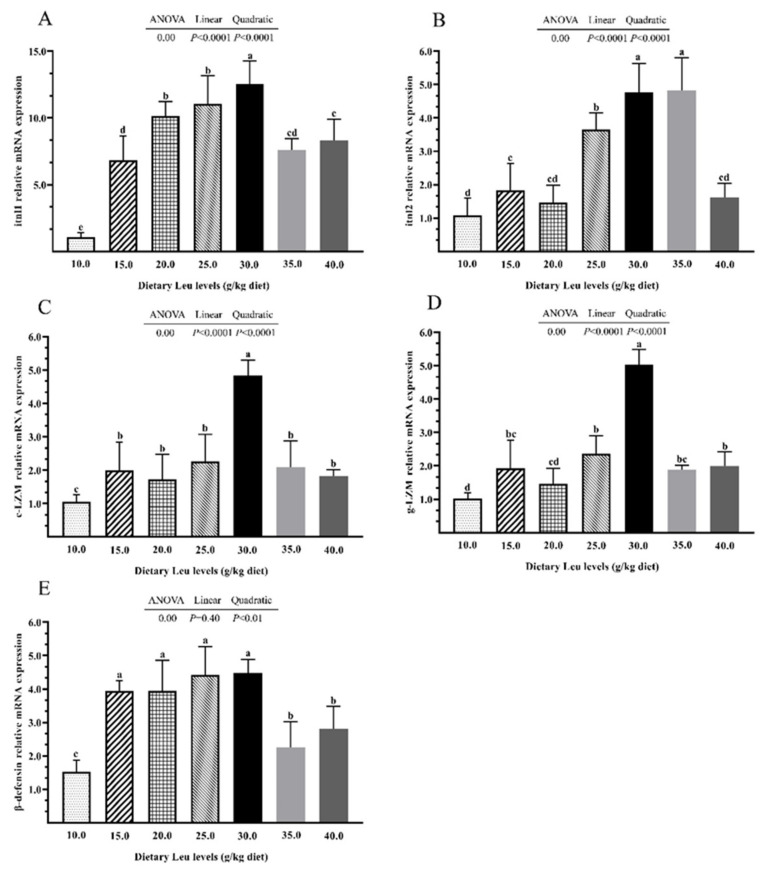
Effects of dietary Leu on itnl1, itnl2, c-LZM, g-LZM, and β-defensin mRNA expressions in intestine of hybrid catfish fed diets with graded levels of Leu (g/kg) for 56 days. (**A**–**E**) The mRNA expressions of itnl1, itnl2, c-LZM, g-LZM, and β-defensin. Results are represented as means ± SEM (n = 3 × 6), with six fish in each replicate. Duncan’s multiple range test was used to detect significance. Different superscripts above the bars indicate significant differences (*p* < 0.05).

**Figure 2 ijms-24-04716-f002:**
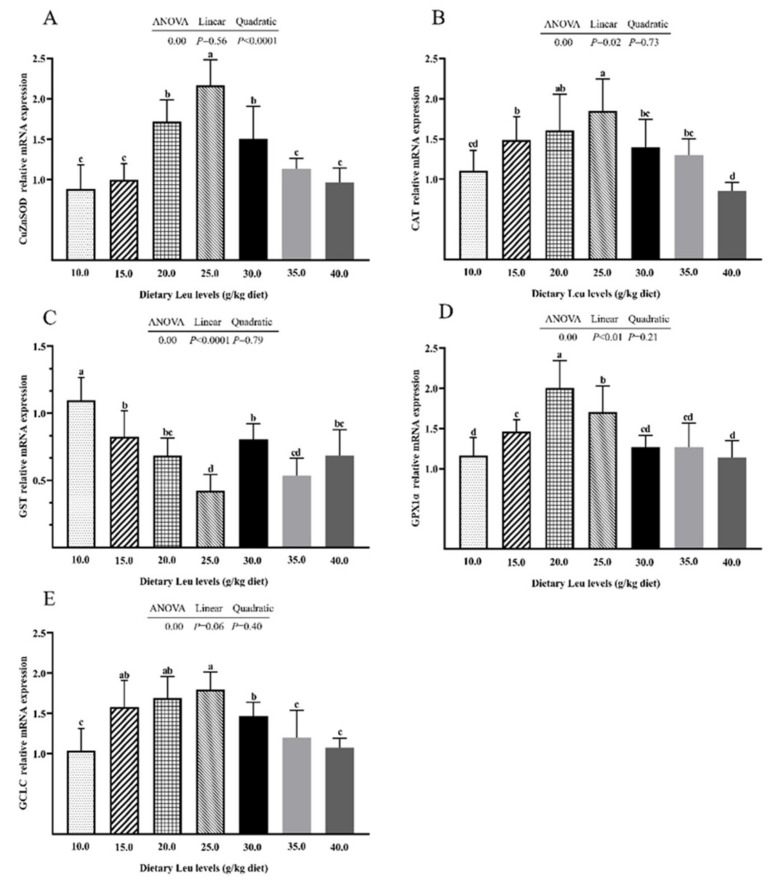
Effect of dietary Leu on the antioxidant capacity in intestine of hybrid catfish fed diets with graded levels of Leu (g/kg) for 56 days. (**A**–**G**) The mRNA expressions of CuZnSOD, CAT, GST, GPX1α, GCLC, Nrf2, and Keap1. (**H**,**I**) The protein levels of Nrf2 and Keap1. Values are means ± SEM (n = 3 × 6), with six fish in each replicate. Duncan’s multiple range test used to detect significance. Different superscripts above the bars indicate significant differences (*p* < 0.05).

**Figure 3 ijms-24-04716-f003:**
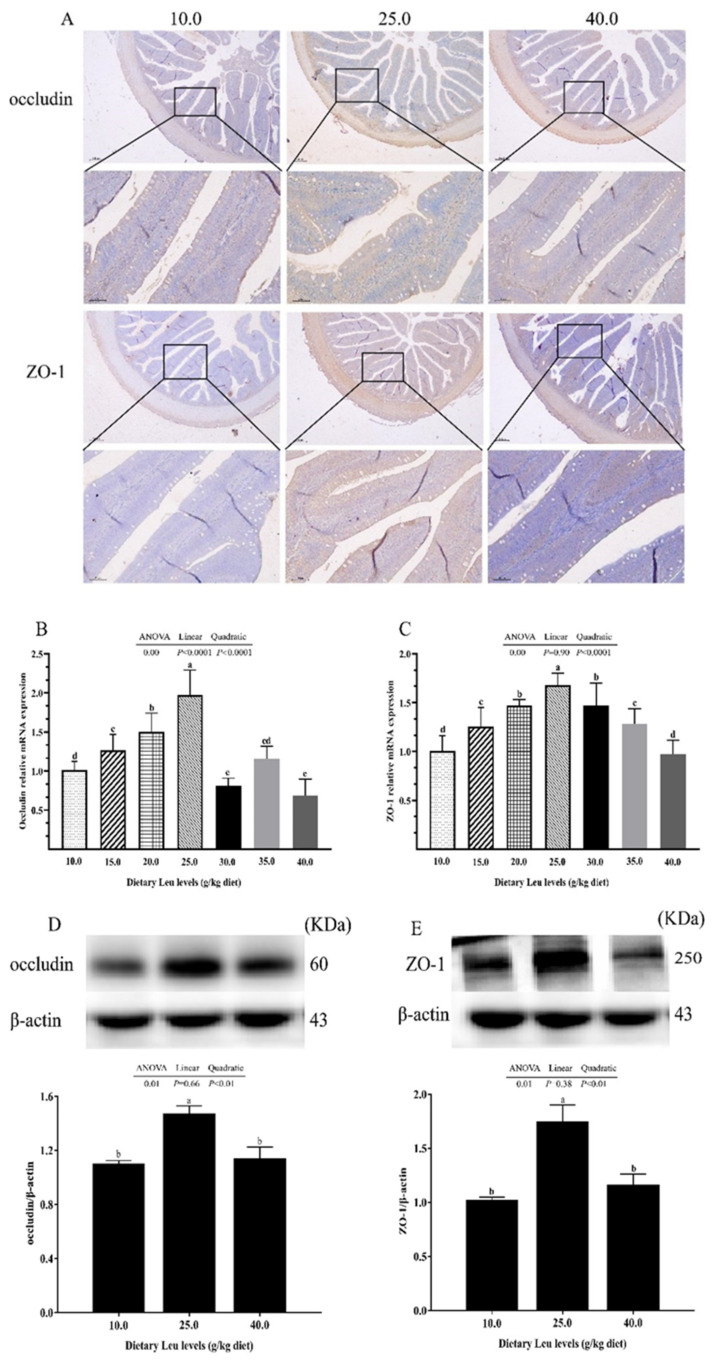
Effect of dietary Leu on TJs in intestine of hybrid catfish fed diets with graded levels of Leu (g/kg) for 56 days. (**A**) IHC staining analysis of occludin and ZO-1 expressions (scale bars: 200 μm and 50 μm). (**B**,**C**,**F**) The mRNA expressions of occludin, ZO-1, and Claudin-2. (**D**,**E**,**G**) The protein levels of occludin, ZO-1, and Claudin-2. All results are represented as means ± SEM (n = 3 × 6). Duncan’s multiple range test was used to detect significance. Different superscripts above the bars indicate significant differences (*p* < 0.05).

**Figure 4 ijms-24-04716-f004:**
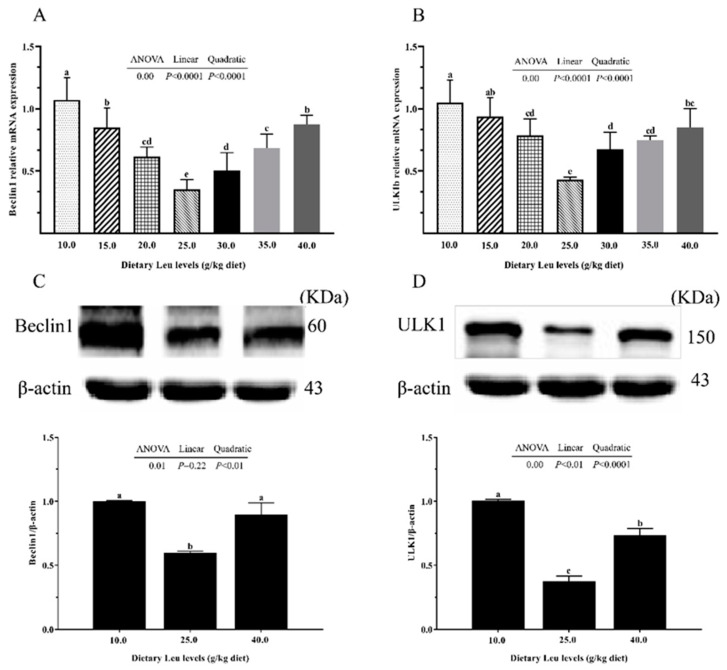
Effect of dietary Leu on autophagy-related gene mRNA expressions and protein levels in intestine of hybrid catfish fed diets with graded levels of Leu (g/kg) for 56 days. (**A**,**B**,**E**–**J**) The mRNA expressions of Beclin1, ULK1b, ATG5, ATG7, ATG9a, ATG4b, LC3b, and P62. (**C**,**D**,**K**,**L**) Protein levels of Beclin1, ULK1, LC3Ⅱ/Ⅰ, and P62. Results are represented as means ± SEM (n = 3 × 6). Duncan’s multiple range test was used to detect significance. Different superscripts above the bars indicate significant differences (*p* < 0.05).

**Figure 5 ijms-24-04716-f005:**
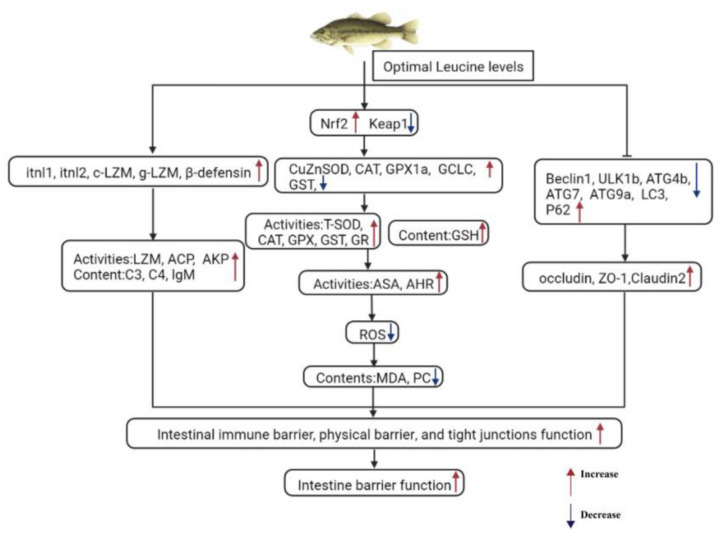
The potential pathways regarding the effects of Leu on the intestinal immune barrier, physical barrier, and tight junction function of fish. Abbreviations: itnl1 = intelectin 1; itnl2 = intelectin 2; c-LZM = c-type lysozyme; g-LZM = g-type lysozyme; Nrf2 = nuclear factor erythroid 2-related factor 2; Keap1 = Kelch-like ECH-associated protein 1; CuZnSOD = copper–zinc superoxide dismutase; CAT = catalase; GPX1a = lutathione peroxidase 1a; GST = glutathione S-transferase; GCLC = catalytic subunit of glutamate–cysteine ligase; ULK1b = uncoordinated 51-like kinase 1b; ATG5 = autophagy-related protein 5; ATG7 = autophagy-related protein 7; ATG9a = autophagy-related protein 9a; ATG4b = autophagy-related protein 4b; LC3b = autophagy marker light-chain 3b; P62 = sequestosome 1; ZO-1 = zonula occludens-1.

**Table 1 ijms-24-04716-t001:** Effects of dietary Leu on immune parameters in the intestine of hybrid catfish fed experimental diets containing graded levels of Leu (g/kg) for 8 weeks ^1^.

Items ^2^	Dietary Leu Levels, g/kg							*Pr* > *F*^2^		
	10.0	15.0	20.0	25.0	30.0	35.0	40.0	ANOVA	Linear	Quadratic
LZM	73.92 ± 3.06 ^c^	87.82 ± 3.23b ^c^	100.07 ± 8.27 ^abc^	117.71 ± 0.36 ^a^	104.7 ± 4.14 ^ab^	85.62 ± 11.13 ^abc^	83.27 ± 3.31 ^c^	0.00	<0.0001	<0.0001
ACP	54.23 ± 2.08 ^e^	77.69 ± 0.63 ^d^	99.98 ± 2.87 ^b^	122.71 ± 5.45 ^a^	100.29 ± 0.50 ^b^	90.72 ± 0.59 ^bc^	86.37 ± 5.42 ^cd^	0.00	<0.0001	<0.0001
AKP	40.75 ± 3.57 ^d^	64.09 ± 4.78 ^c^	83.36 ± 3.33 ^b^	101.82 ± 4.57 ^a^	82.65 ± 1.88 ^b^	69.54 ± 3.84 ^c^	61.94 ± 3.26 ^c^	0.00	<0.0001	<0.0001
C3	9.44 ± 0.46 ^cd^	11.66 ± 0.34 ^b^	14.36 ± 0.66 ^a^	16.02 ± 0.24 ^a^	12.03 ± 0.35 ^b^	10.83 ± 0.56 ^bc^	8.05 ± 1.15 ^d^	0.00	0.02	0.26
C4	0.89 ± 0.04 ^c^	0.9 ± 0.07 ^c^	1.07 ± 0.05 ^bc^	1.38 ± 0.02 ^a^	1.28 ± 0.04 ^ab^	1.21 ± 0.04 ^ab^	0.96 ± 0.14 ^c^	0.00	0.01	0.01
IgM	29.35 ± 0.91 ^d^	32.64 ± 1.33 ^bcd^	38.24 ± 0.77 ^b^	45.05 ± 2.25 ^a^	35.87 ± 1.32 ^bc^	34.15 ± 1.78 ^bcd^	30.21 ± 2.97 ^cd^	0.00	0.74	<0.0001

^1^ Values are mean ± SEM, n = 3. Mean values with different superscripts in the same row are significantly different (*p* < 0.05); ^2^ LZM = lysozyme; ACP = acid phosphatase; AKP = alkaline phosphatase; C = complement; IgM = immunoglobulin M.

**Table 2 ijms-24-04716-t002:** Effects of dietary Leu on antioxidant-related parameters in the intestine of hybrid catfish fed experimental diets containing graded levels of Leu (g/kg) for 8 weeks ^1^.

Items ^2^	Dietary Leu levels, g/kg							*Pr > F* ^2^		
	10.0	15.0	20.0	25.0	30.0	35.0	40.0	ANOVA	Linear	Quadratic
ROS	100.00 ± 7.11 ^a^	83.70 ± 6.12 ^b^	70.10 ± 2.43 ^cd^	52.30 ± 2.60 ^e^	66.80 ± 1.90 ^d^	78.20 ± 1.60 ^bcd^	81.25 ± 8.81 ^bc^	0.00	0.00	0.00
MDA	0.57 ± 0.08 ^a^	0.50 ± 0.07 ^ab^	0.37 ± 0.02 ^b^	0.38 ± 0.10 ^b^	0.41 ± 0.06 ^b^	0.48 ± 0.02 ^ab^	0.51 ± 0.03 ^ab^	0.04	0.03	0.43
PC	2.25 ± 0.17 ^a^	1.86 ± 0.05 ^ab^	1.25 ± 0.26 ^d^	1.46 ± 0.01 ^cd^	1.62 ± 0.05 ^bc^	1.70 ± 0.07 ^bc^	1.91 ± 0.03 ^ab^	0.00	0.04	0.13
GSH	2.16 ± 0.06 ^e^	2.47 ± 0.05 ^cd^	2.64 ± 0.09 ^bc^	2.87 ± 0.10 ^a^	2.67 ± 0.01 ^b^	2.51 ± 0.05 ^bcd^	2.31 ± 0.11 ^de^	0.00	0.05	<0.0001
ASA	184.16 ± 17.73 ^e^	208.87 ± 9.42 ^de^	262.87 ± 21.40 ^b^	293.15 ± 10.29 ^a^	242.93 ± 4.34 ^bc^	218.14 ± 10.42 ^cd^	209.52 ± 5.76 ^de^	0.00	0.13	0.00
AHR	103.79 ± 2.65 ^e^	113.86 ± 4.30 ^cde^	126.45 ± 1.81 ^b^	149.16 ± 8.96 ^a^	123.88 ± 3.26 ^bc^	117.19 ± 1.13 ^bcd^	112.18 ± 2.62 ^de^	0.00	0.08	0.00
T-SOD	26.12 ± 1.30 ^d^	29.23 ± 0.50 ^cd^	32.71 ± 1.30 ^ab^	36.91 ± 1.60 ^a^	31.42 ± 0.70 ^bc^	28.54 ± 1.00 ^c^	27.34 ± 0.90 ^cd^	0.00	0.70	<0.0001
CAT	3.05 ± 0.04 ^d^	3.32 ± 0.08 ^cd^	3.93 ± 0.14 ^ab^	4.48 ± 0.43 ^a^	3.69 ± 0.22 ^bc^	3.37 ± 0.23 ^bcd^	3.18 ± 0.110 ^cd^	0.01	0.68	0.06
GPX	8.05 ± 1.22 ^d^	11.31 ± 0.36 ^bc^	13.08 ± 1.43 ^ab^	15.50 ± 0.90 ^a^	13.31 ± 1.42 ^ab^	12.57 ± 0.62 ^ab^	8.88 ± 0.42 ^cd^	0.00	0.14	0.85
GST	42.68 ± 2.57 ^e^	54.26 ± 4.51 ^cd^	60.65 ± 1.11 ^ab^	63.44 ± 0.10 ^a^	57.76 ± 1.70 ^abc^	55.49 ± 1.07 ^bcd^	50.85 ± 0.13 ^d^	0.00	0.01	0.67
GR	16.79 ± 0.11 ^c^	19.49 ± 1.22 ^b^	21.22 ± 0.22 ^b^	24.22 ± 0.38 ^a^	21.36 ± 0.52 ^b^	19.88 ± 0.56 ^b^	17.47 ± 1.16 ^c^	0.00	0.29	<0.0001

^1^ Values are mean ± SEM, n = 3. Mean values with different superscripts in the same row are significantly different (*p* < 0.05); ^2^ ROS = reactive oxygen species; MDA = malondialdehyde; PC = protein carbonyl; GSH = glutathione content; ASA = antisuperoxide anion; AHR = antihydroxyl radical; T-SOD = total superoxide dismutase; CAT = catalase; GST = glutathione S-transferase; GPX = glutathione peroxidase; GR = glutathione reductase.

**Table 3 ijms-24-04716-t003:** Correlation analysis of parameters in the muscle of hybrid catfish.

Independent Parameters	Dependent Parameters	Correlation Coefficients	*p*
ROS	MDA	0.912	0.004
	PC	0.793	0.033
	ASA	−0.956	0.001
	AHR	−0.954	0.001
	T-SOD	−0.933	0.002
	CAT	−0.928	0.003
	GPX	−0.920	0.003
	GST	−0.943	0.001
	GR	−0.942	0.001
	GSH	−0.965	0.000
CuZnSOD mRNA	T-SOD	0.981	0.000
CAT mRNA	CAT	0.861	0.013
GPX1α mRNA	GPX	0.643	0.119
GST mRNA	GST	−0.792	0.034
Nrf2 mRNA	CuZnSOD mRNA	0.892	0.007
	CAT mRNA	0.794	0.033
	GPX1α mRNA	0.849	0.016
	GCLC mRNA	0.818	0.025
Keap1 mRNA	CuZnSOD mRNA	−0.975	0.000
	CAT mRNA	−0.906	0.005
	GPX1α mRNA	−0.794	0.033
	GCLC mRNA	−0.919	0.003

## Data Availability

The data presented in this study are available in this manuscript.

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
