# Peer review of "Dietary Leucine Improves Fish Intestinal Barrier Function by Increasing Humoral Immunity, Antioxidant Capacity, and Tight Junction"

_ijms, 2023, doi:10.3390/ijms24054716_

Round 1

Reviewer 1 Report

Article no. 2044066

Title: Dietary leucine improves fish intestinal barrier function by in[1]creasing humoral immunity, antioxidant capacity, and tight junction protein occludin, ZO-1, and Claudin-2 levels

Journal: Int. J. Mol. Sci.

Line 18: “105 hybrid catfish” fed with six diets, but mention each diet many fish used?

Line 21: that revealed that the, change appropriately

Line 26: No significant difference on the CAT and GPX activities were detected among treatments (P > 0.05). The CuZnSOD, CAT, and GPX1α mRNA expressions had positive linear; here CAT no significant activity, next sentence mention positive linear, how?

Line 44: What are immunologic and physical barriers? Mention here.

Line 49: Leu requirement of hybrid catfish, incomplete.., for growth?

Line 52: The following references were include after “lysozyme (LZM)”: & Line 252: after “response in fish [34,35]”

doi.org/10.1016/j.vetpar.2010.01.046;

doi.10.1016/j.fsi.2010.02.022;

doi.10.1016/j.fsi.2010.07.006;

doi.10.1577/H09-040.1;

doi.10.1016/j.fsi.2010.08.017

Line 54: The following references were include after “immuno globulin M (IgM)”: Line 256: after “immunity [38]”:

doi.org/10.1016/j.fsi.2018.12.051;

https://doi.org/10.1016/j.fsi.2019.10.033;

https://doi.org/10.1016/j.fsi.2020.02.037;

https://doi.org/10.1038/s41598-020-79629-9;

https://doi.org/10.1016/j.fsi.2021.01.011;

https://doi.org/10.1016/j.fsi.2021.04.025  ;

https://doi.org/10.1016/j.fsi.2021.07.009

Line 54: The following references were include after “environment [13,14]”: Line 279: after “of ROS [49]”: Line 298: after “of ROS [49]”

https://doi.org/10.1016/j.fsi.2021.08.004;

https://doi.org/10.1016/j.fsi.2021.09.024.;

https://doi.org/10.1016/j.fsi.2021.09.040;

https://doi.org/10.1016/j.fsi.2021.10.026;

https://doi.org/10.1016/j.fsi.2021.12.033;

https://doi.org/10.1016/j.fsi.2022.01.022

Line 79: Scientific name include after grass carp..

Line 97: Usually with test material as control But this study 10.0 g/kg Leu diet used as control group??? How??

Line 98: totally how many fish used in this study?

Line 109: (MOSS and Allam, 2007), reference number only provide here..

Line 130: Superscript of Script®

Line 165: Italics of P < 0.05

Line 176: Which Leu level better humoral immune activity as mention in last sentence…

Line 195: Here mention which Antioxidant best of the Leu diets???

Line 209: A conclusion sentence was including here…

Line 228: What about the effect of Luc on TJs proteins and mention a sentence..

Line 260: Here mention which dietary Leu levels better or significant LZM, ACP, and AKP and C3, C4 activities.

Line 264: mention the scientific name of the fish here;

Line 268: Here mention which dietary Leu levels better or significant itnl1, itnl2, c-LZM, g-LZM, and β-defensin expression. Here include any fish work on Leu

Line 281: why decreased ROS, PC and MDA contents after Leu treatment

Line 264: Include optimal Leu level on intestinal antioxidant activity

Line 288: mention fish name here.

Line 294: Here mention other amino acid in different fish on antioxidant activity..

Line 311: positive linear to modify significantly /in significantly…appropriately

Line 316: Include scientif name of Jian carp’’’

Line 330: un italic of fingerlings

Line 343: increased linearly is increased significantly?

Line 324: to be italic of In vitro

Line 351: optimal Leu level include here

Line 368: linearly and/or quadratically decreased is decrease significantly?

Reviewer 2 Report

This manuscript is about the use of dietary leucine for improving fish intestinal barrier function by increasing humoral immunity, antioxidant capacity and tight junction protein.

The manuscript is well written and presented, and the results are interesting. 

However, I have some remarks: 

Abstract 

Line 21: delate , showed or that revealed 

Line 147-Line 166 add reference 

Table 1: add references for primer sequences 

Discussion: Is there any relation between leucine and microbiota? . If there is, the authors should discuss the impact or the relation. 

Conclusion: The authors should add perspectives 

Reviewer 3 Report

This study investigated the effect of dietary leucine on intestinal tight junction barrier in fish by suppressing oxidative stress and enhancing intestinal tight junction integrity. This manuscript is interesting but can be improved. In addition, there are several critical concerns as mentioned below.

- The authors measured intestinal tight junction-dependent physical barrier by investigating mRNA and protein expression using qRT-PCR and western blot, respectively. In my opinion, only qRT-PCR and western blot are not enough to inform about tight junction-intestinal barrier function since protein expression do not recapitulate tight junction integrity. Therefore, immunofluorescence staining of tight junction protein localization should be performed. If you are not able to search for antibodies used for immunofluorescence that recognize tight junction protein in fish, transmission electron microscope can be used instead. 

- Moreover, permeability assay should also be performed to determine tight junction-dependent intestinal barrier function since there are also several probes used for determination of both tight junction-dependent and tight junction-independent paracellular permeability already available. 

- There is no experiment that clearly proves the pathway shown in Figure 5.  

Reviewer 4 Report

This paper focuses on a hybrid catfish. This model seems to be used only by this team. Please add more references to highlight this model's interest for other future researchers and what has been done by different groups. Without this, it's not suitable for a broad readership like in IJMS.

The same team has already published a paper but with another amino acid, threonine. By changing the amino acid, the scientific input is very low. A comparison between more amino acids would be more interesting for the community.

Metabolically speaking, did the authors have any ideas about how leucine could impact immune response in animals?

Figure 1 is not easy t follow. What a,b,c, and d are meaning? Please explain in the legend below.

For all these reasons I recommend a major revision.

Reviewer 5 Report

The authors describe important findings from an investigation of dietary modification on fish intestinal immune function. The results are clearly presented in the tables and figures.

Note that this reviewer does not investigate fish and therefore cannot comment on fish care/treatment described in this manuscript.

Please edit the manuscript for proper English sentence structure and grammar.

Abstract: Please summarize the findings more succinctly and define the abbreviations.

Line 111: Six-μm thick sections

247: immune barrier

250: choose a word other than "ponderable."

A discussion of the implications of your results on fish growth rate, maximum growth and market supply would be helpful.

Figure 5: This is a valuable figure for summarizing your interpretation of the findings. Include definitions of the abbreviations in the footnote.

Table S2: Tryptophan is not essential for fish?

Round 2

Reviewer 3 Report

Please re-check English usage in the current form of manuscript?

Reviewer 4 Report

My main remarks are not taken into account in the manuscript.

The authors need to state clearly in the introduction that their model is used by different teams or even better, in the food market. 

Leucine is an essential amino acid like many others... The authors need to make modifications to the main text to help the readership understand the value of their work.

Round 3

Reviewer 4 Report

The authors have modified the manuscript as requested.